# POPI: Personalizing LLMs via Optimized Natural Language Preference Inference

## Abstract

Large language models (LLMs) are typically aligned with population-level preferences, despite substantial variation across individual users. We introduce POPI, a user-level personalization framework that separates the problem into two components connected by a natural-language interface: a shared inference model that distills heterogeneous user signals into a concise preference summary, and a shared generator that conditions on this summary to produce personalized responses. Both components are trained under a unified preference-optimization objective, with reinforcement learning handling the non-differentiable inference step. This objective decomposes into generator approximation error and summary informativeness, revealing how a single loss simultaneously drives accurate generation and informative summarization. Because the interface is natural language, learned summaries can be inferred once per user and reused across different generators—including frozen, black-box commercial APIs. Across four personalization benchmarks, POPI generally improves personalization quality while reducing context overhead by up to an order of magnitude.

## 1 Introduction

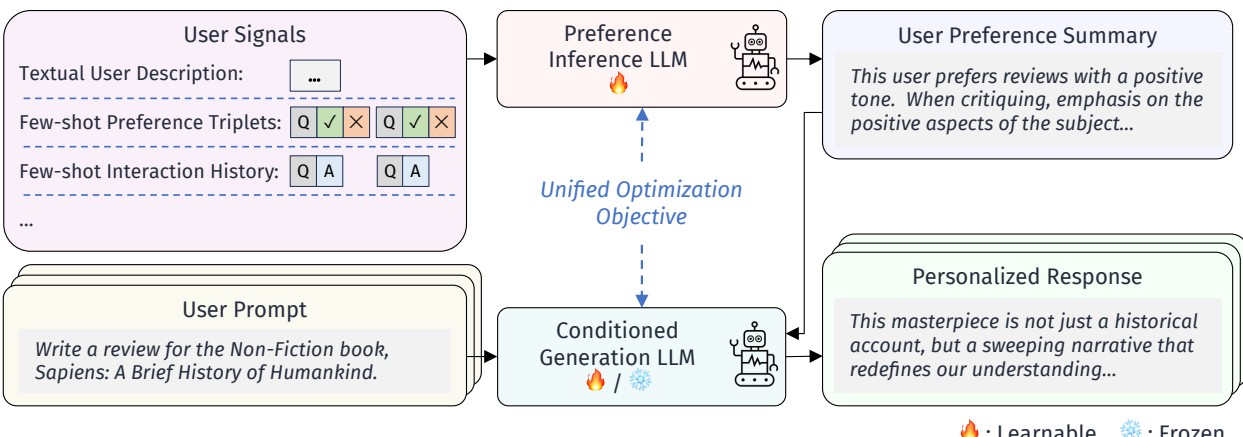

Figure 1: **Inference-time usage of POPI.** POPI instantiates the advocated modular design for user-level personalization. A shared inference model distills heterogeneous, sparse user signals into a concise natural-language preference summary, which serves as a generator-agnostic interface. A shared generator then conditions on both the task prompt and this summary to produce personalized responses. Crucially, this modular design allows inferred summaries to be reused with other frozen, off-the-shelf LLMs, including black-box commercial APIs.

Large language models (LLMs) have rapidly become integral tools for information access (Zhu et al., 2023), creativity (Franceschelli & Musolesi, 2025), and decision support (Vrdoljak et al., 2025). Despite strong aggregate performance, user satisfaction is often inconsistent: individuals may strongly prefer one model or

response style over another even when benchmark results are similar (Li et al., 2024; Guan et al., 2025). Such divergence underscores a key reality: **the "best" model is not universal but inherently user-dependent**. The resulting challenge is not only to improve general capability but also to scalably adapt LLM behavior to diverse individual preferences.

Despite growing interest, effective LLM personalization remains technically challenging. Fine-tuning a separate model per user with alignment methods such as RLHF (Ouyang et al., 2022) or DPO (Rafailov et al., 2023) is computationally infeasible. Recent work has therefore shifted toward shared-model approaches that condition a single generator on user-specific signals—personas, interaction histories, or preference-labeled examples, which are often scarce, heterogeneous, and noisy, making it nontrivial to extract stable preference information (Singh et al., 2025; Zhao et al., 2023a; 2025; Li et al., 2025; Lee et al., 2024a; Zhou et al., 2024b).

In this paper, we focus on **user-level personalization**, where a user's core preferences are assumed to be relatively stable across interactions. This setting has an important practical advantage: preferences need only be inferred once per user and can then be reused across many downstream prompts, making it well-suited for scalable deployment.

User-level personalization methods vary widely in how they incorporate user signals, but they can be usefully viewed through a common lens: they decompose into a **preference inference** step that extracts stable user information from noisy signals, and a **conditioned generation** step that uses this information to produce personalized responses. In Section 3, we make this structure explicit and argue for a modular treatment in which these two components are separated and connected by a natural-language interface. This modular design allows the two components to be developed and improved independently by the community, and encourages a practical benefit: **generator-transferability**, where a preference summary inferred once can be reused with different generators—including black-box APIs—without retraining.

Guided by this abstraction, we propose POPI (**P**ersonalizing LLMs via **O**ptimized Natural Language **P**reference **I**nference), a concrete instantiation of modular user-level personalization. Figure 1 illustrates the inference-time usage of POPI. POPI employs a preference inference model to distill heterogeneous user signals into concise natural-language preference summaries, which serve as the sole interface to a generator. Both components are shared by all users and are trained under a **unified objective**, with reinforcement learning serving as an optimization tool.

We evaluate POPI on four benchmarks for LLM personalization, spanning sentiment analysis, explanation generation, open-ended dialogue, and forum discussion. Across these settings, POPI generally outperforms shared-model baselines in the non-fine-tuning regime, where optimized summaries applied to a frozen generator yield noticeable win-rate improvements over unoptimized alternatives while reducing context overhead by up to an order of magnitude. In generator-transferability experiments, the learned summaries generally transfer well to a diverse set of frozen generators—including black-box commercial APIs—lending empirical support to the modular design.

In summary, this work makes the following contributions: **(1)** We introduce POPI, a framework that separates user-level personalization into a preference-inference model and a conditioned generator connected by a natural-language interface, and jointly trains both components under a unified preference-optimization objective. **(2)** We show that this objective admits an information-theoretic decomposition into generator approximation error and summary informativeness, making precise how the unified objective simultaneously trains an accurate generator and an informative summarizer. **(3)** Across four benchmarks, we show that POPI generally improves personalization quality while reducing context overhead by up to an order of magnitude. **(4)** We demonstrate that the learned summaries transfer to frozen, off-the-shelf generators—including black-box commercial APIs. We release our implementation at `https://anonymous.4open.science/r/POPI-384F/`.

## 2 Related Work

**User Profiling outside LLM Alignment.** User modeling has a long history in NLP and related fields, spanning personalized query reformulation (Zhang, 2022; Hui et al., 2024), summarization (Zhang et al., 2024; Ghodratnama & Zakershahrak, 2024), and authorship attribution (He et al., 2024; Deutsch & Paraboni, 2023). In recommender systems, preference inference from implicit or explicit signals has long been central,

from classical matrix factorization and collaborative filtering (Koren et al., 2009; Sarwar et al., 2001) to recent approaches that represent user profiles directly in natural language (Ramos et al., 2024; Zhou et al., 2024a; Gao et al., 2024; Radlinski et al., 2022). Related profiling techniques also appear in market research (Yin et al., 2025; Rizwan et al., 2025) and adaptive education (Liu et al., 2025; 2024). Our work shares both the motivation of extracting preferences from heterogeneous signals and, in particular, the choice of natural language as a user-profile representation (Ramos et al., 2024; Radlinski et al., 2022; Zhou et al., 2024a). The main difference is that we optimize the natural-language summary end-to-end against a downstream preference-alignment objective, rather than treating it as a preprocessing step or a recommendation representation.

**Preference Alignment Methods for Average User.** Preference alignment methods aim to adapt LLMs to population-level preferences (Wang et al., 2023). RLHF remains the standard approach, using reward models trained on human judgments to guide policy optimization (Christiano et al., 2017; Ouyang et al., 2022), with extensions that incorporate AI feedback (Bai et al., 2022). DPO (Rafailov et al., 2023) removes explicit reward modeling, and variants such as ORPO (Hong et al., 2024) and SimPO (Meng et al., 2024) further simplify optimization. Other lines reshape the loss (Ethayarajh et al., 2024), enforce ranked log-probability consistency (Yuan et al., 2023), or calibrate likelihoods to preferences (Zhao et al., 2023b). While effective for aligning to the "average" user, these methods largely overlook individual variation, motivating growing interest in personalized alignment (Xie et al., 2025).

**Personalized LLM Alignment.** Early personalization approaches adapt LLM behavior by learning user-specific adapters or embeddings, such as HyRe (Lee et al., 2024b), LoRe (Bose et al., 2025), PAL (Chen et al., 2024), and VPL (Poddar et al., 2024). These methods typically operate at the reward modeling level and require per-user optimization, limiting scalability. To avoid per-user training, subsequent work adopts a meta-learning paradigm in which a shared generation model conditions on user signals provided as additional context. Representative examples include GPO (Zhao et al., 2023a), FSPO (Singh et al., 2025), and NextQuill (Zhao et al., 2025), which differ in optimization objectives but share a reliance on raw or lightly processed user signals. While effective, such approaches often incur substantial context overhead and are sensitive to noisy or heterogeneous inputs. More structured methods distill user signals into explicit profiles before conditioning generation, including handcrafted persona extraction (Lee et al., 2024a), projection into predefined preference dimensions (Li et al., 2025), and hypothesis-based user descriptions (Garbacea & Tan, 2025; Zhou et al., 2024b). Although these approaches improve interpretability, they typically treat preference inference as a static or heuristic preprocessing step. In contrast, POPI is **complementary** to this line of work: manually constructed profiles can serve as inputs to our preference inference model, which further refines them through unified optimization to produce summaries directly consumable by the generation model.

## 3 Abstraction of User-level Personalization

**User-level personalization**, as the term is commonly used in the literature, refers to the task of adapting a model's generation behavior to a user's preferences, given sparse and heterogeneous user signals, under the assumption that those preferences are relatively stable across interactions and independent of the specific upcoming prompt. Preferences do evolve over time, but in many practical applications, such as personalized content generation, education, and dialogue, they remain consistent over the relevant time horizon. This stability also enables an important efficiency benefit: preferences can be inferred once and reused across many prompts, amortizing the cost of preference extraction. Many algorithms have been developed in this setting; the rest of this section makes the structure they share explicit, decomposing user-level personalization into two components, clarifying how they interact, and advocating for a modular design principle that supports generator-transferability.

### 3.1 Problem Formulation and Modular Decomposition

We consider a population of $N$ users, indexed by $i$. For each user, the available supervision takes the form of a dataset of preference-labeled examples,

$$\mathcal{D}_i = \{(x, y_c, y_r)\},$$

where $x$ is a prompt and $y_c, y_r$ denote the preferred and dispreferred responses, respectively. In addition, each user is associated with a set of user-specific signals carrying information about their preferences, such as personas, interaction histories, or a small number of labeled examples, which we collectively denote by $s_i$.

At an abstract level, user-level personalization methods can generally be decomposed into two components: **(1)** a **preference inference** function $f$ that maps the user signals $s_i$ to a representation of the user's preferences,

$$z_i = f(s_i),$$

and **(2)** a **conditioned generation** function $g$ that uses this representation, together with the prompt $x$, to produce a personalized response,

$$y = g(x, z_i).$$

Existing methods can be viewed as instantiations of this structure, either explicitly or implicitly, differing primarily in how $f$ and $g$ are parameterized and trained using the per-user supervision $\mathcal{D}_i$.

Making this decomposition explicit reveals a natural opportunity for **modular design**: treating preference inference and conditioned generation as separate components connected by a well-defined interface. We adopt this modular design principle for two reasons. First, it lets the two components be developed and improved independently. Second, it allows an inferred summary $z_i$ to be paired with any generator—including models with distinct architectures or those accessed only via black-box APIs—without retraining the inference model. We refer to this second property as **generator-transferability**. It is practically valuable in deployment settings where the generator may be swapped, upgraded, or selected per query. Prior personalization work has largely under-explored this property, and we evaluate it explicitly in our experiments.

### 3.2 Natural Language as a Generator-Agnostic Interface

Given the decomposition above, the intermediate representation $z_i$ is the sole interface between preference inference and conditioned generation. If these two components are to remain meaningfully modular rather than tightly coupled, the format of this interface must be independent of any particular generator, while remaining expressive enough to capture diverse user preferences.

Natural language (with optional lightweight formatting, e.g., Markdown) is well-suited to this role. As the native input modality of modern LLMs, it is directly consumable by any model without architectural modification or fine-tuning, yet retains the expressivity needed for nuanced preferences. Latent vector representations, by contrast, tie preference inference to a specific generator; transferring them to other generators typically requires additional fine-tuning or semantic alignment.

## 4 Method

We now instantiate the abstraction of Section 3 and introduce POPI, a personalization framework that explicitly supports the modular design principle, uses natural language as the interface, and jointly trains preference inference and conditioned generation under a unified objective.

Following the decomposition, we adopt a simple parameterization for both components: preference inference $f$ is implemented as a language model $\pi_\phi$ that maps user-specific signals $s_i$ to a natural-language preference summary $z_i$,

$$z_i \sim \pi_\phi(\cdot \mid s_i),$$

while conditioned generation $g$ is implemented by another language model $\pi_\theta$ that conditions on both the task prompt $x$ and the summary $z_i$ to produce a response,

$$y \sim \pi_\theta(\cdot \mid x, z_i).$$

Both models are shared across all users. An overview of the resulting training framework is shown in Figure 2.

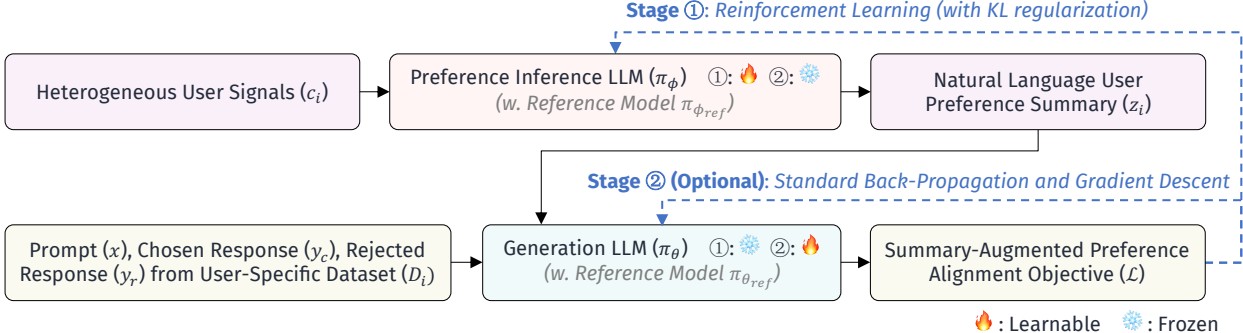

Figure 2: Training framework of POPI. The preference inference model $\pi_\phi$ maps raw user signals to a natural-language preference summary, which serves as the interface between inference and generation. Together with preference-labeled data, this summary is used to define a summary-augmented preference optimization objective that provides a unified training objective for both models. Training proceeds in two stages: $\pi_\phi$ is optimized with reinforcement learning, and then $\pi_\theta$ is optionally fine-tuned via standard backpropagation.

### 4.1 Unified Optimization Objective for Preference Inference and Generation

When user-specific preference-labeled data $\mathcal{D}_i$ is available, it is natural to use this supervision to improve both the preference inference model $\pi_\phi$ and the generation model $\pi_\theta$, rather than treating either as a fixed oracle.

**Summary-Augmented Preference Alignment Objective.** We first derive a training objective for the generation model $\pi_\theta$ by adapting existing average-user preference alignment algorithms. The same adaptation procedure applies broadly across different algorithms; as a concrete illustration, we use Direct Preference Optimization (DPO) (Rafailov et al., 2023) below. Generalization to other alignment methods is discussed in Section 4.3.

If we could afford to train a dedicated generator $\widetilde{\pi}_{\theta_i}$ for each user using $\mathcal{D}_i$, then under DPO, the per-user per-example objective would be

$$\widetilde{\ell}_i = -\log\sigma\left(\beta\log\frac{\widetilde{\pi}_{\theta_i}(y_c \mid x)}{\pi_{\theta_{\mathrm{ref}}}(y_c \mid x)} - \beta\log\frac{\widetilde{\pi}_{\theta_i}(y_r \mid x)}{\pi_{\theta_{\mathrm{ref}}}(y_r \mid x)}\right),$$

where $\pi_{\theta_{\mathrm{ref}}}$ is a fixed reference model. Averaging over all examples and users, the aggregated loss would be

$$\widetilde{\mathcal{L}} = \frac{1}{N}\sum_{i=1}^{N}\mathbb{E}_{(x,y_c,y_r)\sim\mathcal{D}_i}\left[\widetilde{\ell}_i\right].$$

In practice, however, training a separate generator per user is infeasible. Following the decomposition introduced earlier, we therefore replace $\widetilde{\pi}_{\theta_i}(y \mid x)$ with the shared generator $\pi_\theta(y \mid x, z_i)$ that conditions on the user-specific natural-language summary $z_i$. This yields the **summary-augmented** DPO objective:

$$\ell_i = -\log\sigma\left(\beta\log\frac{\pi_\theta(y_c \mid x, z_i)}{\pi_{\theta_{\mathrm{ref}}}(y_c \mid x)} - \beta\log\frac{\pi_\theta(y_r \mid x, z_i)}{\pi_{\theta_{\mathrm{ref}}}(y_r \mid x)}\right),$$

$$\mathcal{L} = \frac{1}{N}\sum_{i=1}^{N}\mathbb{E}_{(x,y_c,y_r)\sim\mathcal{D}_i,\ z_i\sim\pi_\phi(\cdot\mid s_i)}[\ell_i],$$

$$(1)$$

which can be used to supervise the generation model $\pi_\theta$ via standard back-propagation. Note that $\mathcal{L}$ treats $z_i$ as given; we now turn to how the inference model that produces $z_i$ is itself trained.

**Supervising Preference Inference via the Same Objective.** The summary-augmented objective $\mathcal{L}$ does more than train the generator: it can *naturally serve as the training objective for the preference inference*

*model* $\pi_\phi$ as well. Intuitively, a preference summary $z_i$ receives a lower loss when conditioning on it helps $\pi_\theta$ better distinguish the preferred response from the rejected one.

The objective $\mathcal{L}$ is, however, **not directly differentiable** with respect to the parameters of $\pi_\phi$, because the preference summary is obtained by **sampling** $z_i \sim \pi_\phi(\cdot \mid s_i)$. Rather than modifying the objective, we treat this as a standard problem of optimizing a language model against a non-differentiable downstream loss. Any policy-gradient-based method could handle this in principle; we adopt **reinforcement learning**, specifically Group Relative Policy Optimization (GRPO) (Shao et al., 2024), for its efficiency and stability in fine-tuning language models (Zhang et al., 2025; Mroueh et al., 2025).

Concretely, we view $\pi_\phi$ as a stochastic policy whose actions are natural-language preference summaries. For each user signal $s_i$, we sample a group of summaries $z_i \sim \pi_\phi(\cdot \mid s_i)$ and assign each a scalar reward equal to the negative summary-augmented objective $-\ell_i$. GRPO then optimizes $\pi_\phi$ by encouraging summaries that achieve higher relative reward within the group, so that maximizing expected reward is exactly equivalent to minimizing $\mathcal{L}$.

To further stabilize training and preserve the interpretability of summaries, we include a KL divergence regularization term that anchors $\pi_\phi$ to a fixed reference model $\pi_{\phi_{\mathrm{ref}}}$, yielding the final objective:

$$\mathcal{L} + \alpha \, \mathrm{KL}(\pi_\phi(\cdot \mid s_i) \, \| \, \pi_{\phi_{\mathrm{ref}}}(\cdot \mid s_i)).$$

Empirically, we observe that training with this regularized objective tends to yield **shorter summaries**, even though no length penalty appears in the objective (see Section 5.2 for a quantitative analysis). One plausible explanation is that, because only preference-predictive content contributes positively to the reward while the KL term discourages free-form deviation from the reference policy, the inference model has little incentive to retain material that is not useful for the downstream preference task, therefore producing a compression effect. This emergent behavior is practically beneficial in the personalization setting, where user signals are typically verbose and sparse, and it spares us from introducing explicit length-based constraints.

## 4.2 Two-Stage Training

To preserve modularity, we adopt a two-stage training procedure rather than optimizing both models simultaneously. In Stage 1, we optimize the inference model $\pi_\phi$ using GRPO as described above, while keeping the generator $\pi_\theta$ fixed. In an **optional** Stage 2, we fine-tune $\pi_\theta$ with standard backpropagation using the same objective, conditioning on the learned summaries. Algorithms 1 and 2 in Appendix A summarize the training and inference procedures.

This design involves a trade-off. By fixing the generator in Stage 1, the inference model is encouraged to produce summaries that are informative on their own and broadly useful across generators, yielding the transferability demonstrated in our experiments. However, fixing the generator may also cause the inference model to exclude information that the base generator finds difficult to exploit but that a subsequently fine-tuned generator could have leveraged, potentially limiting Stage 2 gains. We view this as a choice favoring generality and transferability, and discuss it further in the later sections.

## 4.3 Generalization Across Alignment Frameworks

Although we instantiate POPI using DPO in Section 4.1, the framework itself can be generalized to other preference alignment algorithms, by conditioning the generator on the interface $z_i$ without altering their core structure. As another concrete illustration, we consider Identity Preference Optimization (IPO) (Azar et al., 2024). Following the same adaptation procedure as in Section 4.1, we replace the dedicated per-user generator with the shared generator $\pi_\theta(y \mid x, z_i)$, yielding the **summary-augmented** IPO objective:

$$\ell'_{\mathrm{IPO},i} = \left( \log \frac{\pi_\theta(y_c \mid x, z_i)}{\pi_{\theta_{\mathrm{ref}}}(y_c \mid x)} - \log \frac{\pi_\theta(y_r \mid x, z_i)}{\pi_{\theta_{\mathrm{ref}}}(y_r \mid x)} - \frac{1}{2\beta} \right)^2,$$

$$\mathcal{L}'_{\mathrm{IPO}} = \frac{1}{N} \sum_{i=1}^{N} \mathbb{E}_{(x,y_c,y_r)\sim\mathcal{D}_i, \ z_i\sim\pi_\phi(\cdot|s_i)} \left[ \ell'_{\mathrm{IPO},i} \right].$$

$$(2)$$

Empirically, our ablation studies show that summary-augmented IPO achieves performance comparable to its DPO counterpart, suggesting that the framework is not tightly coupled to a specific alignment algorithm.

### 4.4 A Prompt-Dependent Variant

POPI as described above targets the user-level setting, where preferences are assumed to be relatively stable across prompts. The framework nonetheless admits a natural extension beyond this scope: the preference summary $z_i$ can be additionally conditioned on the upcoming prompt $x$,

$$z_i \sim \pi_\phi(\cdot \mid s_i, x),$$

yielding a per-prompt summary rather than a per-user one. Because the summary must be recomputed for each new prompt, this variant falls outside the user-level scope and forfeits the modularity and reusability benefits discussed in Section 3. We study it as an ablation in Section 5.4. Empirically, its gains over the prompt-independent formulation are limited, so we adopt the prompt-independent variant throughout the remaining experiments.

### 4.5 Interpretation: Information-Theoretic View

To verify that $\mathcal{L}$ is aligned with the conceptual goals of our modular design, we examine what it implicitly asks the two models to do. Building on the Bradley–Terry (BT) model of preferences (Bradley & Terry, 1952) and following the analysis of Rafailov et al. (2023), the summary-augmented DPO objective (Equation 1) admits the decomposition

$$\mathcal{L} = \underbrace{\mathrm{KL}(P(y_c \succ y_r \mid x, z_i) \, \| \, P_{\pi_\theta}(y_c \succ y_r \mid x, z_i))}_{\text{generator approximation error}} + C - \underbrace{I(y_c \succ y_r \, ; \, z_i \mid x)}_{\text{summary informativeness}}, \tag{3}$$

where $C = H(P(y_c \succ y_r \mid x))$ is independent of both models' parameters. Minimizing $\mathcal{L}$ therefore simultaneously pursues two effects: it drives the generator $\pi_\theta$ to approximate the true preference distribution under the summary, and it pushes the inference model $\pi_\phi$ to produce summaries carrying preference-relevant information beyond what the prompt $x$ already conveys. This confirms that $\mathcal{L}$ operationalizes the two desiderata of the modular design: an accurate generator and an informative summary. A full derivation is provided in Appendix B.

## 5 Experiments

We evaluate POPI across four benchmarks spanning different personalization settings.

### 5.1 Experimental Setup

**Datasets.** We evaluate POPI on four public benchmarks for personalized preference alignment. The first three datasets (Singh et al., 2025) target distinct personalization settings: `Review` (sentiment and conciseness in media reviews), `ELIX` (explanations across five educational levels in scientific domains), and `Roleplay` (open-ended dialogue conditioned on diverse complex personas). Following prior work, we train on 20k samples per dataset, using only a small number of labeled preference pairs as user signals (four for `Review` and `ELIX`, eight for `Roleplay`) to simulate low-resource personalization. We additionally evaluate on `AlignX` (Li et al., 2025), a large-scale forum discussion dataset. Its user signals include demographics, user-generated content, and preference-labeled pairs. `AlignX` also provides a **handcrafted 90-dimensional preference direction** derived from these signals. We therefore consider two settings: **(1)** using raw user signals as input, or **(2)** using the extracted preference directions as inputs. Following the official protocol, we train on 92k samples (7%) and evaluate on four test splits: `DEMO`, `PAIR`, `UGC`, and `Arbitrary`, isolating different signal sources.

**Baselines.** We organize baselines and our method into two groups based on whether the generation model is fine-tuned. The first group contains methods that do not fine-tune the generator. **(1) Base-Model** conditions generation only on the task prompt. **(2) Raw-Prompting** prepends raw user signals directly to the prompt,

conceptually encompassing Zhou et al. (2024b). **(3) Summary-Prompting** conditions the frozen generator on a natural-language preference summary produced by the unoptimized inference model $\pi_{\phi_{\text{ref}}}$. This baseline serves as a controlled ablation; not as a stand-in for prior handcrafted user profile extraction methods, which are complementary to POPI because their handcrafted user profiles can serve as inputs to our preference inference model (an example is shown in Section 5.3). **(4) POPI-Inference-Only** represents POPI with only training Stage 1. It conditions the frozen generator on POPI's optimized preference summaries. These methods can also be directly applied to personalize off-the-shelf, black-box LLMs.

The second group contains methods that additionally fine-tune the generator using preference-labeled data. **(1) Raw-Aligned** fine-tunes the generator while conditioning on raw user signals, conceptually encompassing Singh et al. (2025); Li et al. (2025); Zhao et al. (2023a; 2025); Lee et al. (2024a). **(2) Summary-Aligned** is the generator-fine-tuning counterpart of Summary-Prompting. **(3–5) LoRe** and **PAL-A / PAL-B** learn **user-specific latent embeddings** that are injected into the generator during fine-tuning. As a result, these methods do not support frozen generators and do not transfer to off-the-shelf generators. **(6) POPI-Full** corresponds to our full method described in Section 4. For all methods, we use the same underlying preference optimization algorithm (DPO, unless otherwise specified) to ensure fair comparison.

**Evaluation Metrics.** We evaluate personalization on **held-out users** using three complementary metrics. **Win Rate with LLM-as-a-Judge (Win Rate).** On `Review`, `ELIX`, and `Roleplay`, each test instance includes a ground-truth persona available only at evaluation time. Following the official evaluation protocol of these datasets (Singh et al., 2025), we use GPT-4o (OpenAI, 2024a) as a judge to compare model outputs and select the response better aligned with the persona (see Appendix J for the prompting template). We further complement Win Rate with a second metric that does not rely on an external judge. **Reward Accuracy (Acc.)** measures alignment with ground-truth preference labels: a prediction is correct if the model assigns a larger log-probability margin between the preferred and rejected responses than the Base-Model does, capturing how much the personalization method shifts the generator toward the correct preference relative to an unpersonalized baseline. **Average Context Overhead (Avg. Len.)** reports the average number of additional tokens introduced by user-specific conditioning, measuring context efficiency.

**Hyperparameters.** In DPO-based experiments, we scale the KL regularization weight $\alpha$ linearly with the DPO temperature $\beta$ to keep the KL and log-ratio terms on a comparable scale. Since larger $\beta$ amplifies the preference log-ratios, a proportionally larger KL weight is required to maintain balance. With this coupling, training becomes empirically robust to the specific choices of $\alpha$, and we therefore fix $\alpha = 0.002 \cdot \beta$ in all experiments. For IPO-based experiments, we follow the same principle and fix $\alpha = 0.002 \cdot \frac{2}{\beta}$. We select $\beta$ via grid search over $\{0.1, 0.05, 0.01\}$. Unless otherwise stated, both $\pi_\phi$ and $\pi_\theta$ are initialized from LLaMA-3.2-3B-Instruct (Dubey et al., 2024) with the default chat template. We use a batch size of 8, a learning rate of $1 \times 10^{-6}$, and a cosine learning rate schedule with 150 warmup steps for both training stages of POPI and all baselines.

**Generator-Transferability to Off-the-Shelf LLMs.** To evaluate generator-transferability, we take the preference summaries learned by POPI and use them to condition a diverse set of frozen, off-the-shelf LLMs that were not involved in training. The set spans both open-weight and black-box commercial models: Mistral-Small (Mistral-S) and Mistral-Large (Mistral-L) (AI, 2024), DeepSeek-R1 (Guo et al., 2025), Claude-4-Sonnet (Claude-4) (Anthropic, 2025), GPT-4o-mini (OpenAI, 2024b), and three LLaMA-3.2-Instruct variants (Llama-1b, Llama-11b, Llama-90b) (Dubey et al., 2024). We use these shorthand names consistently in all result tables.

**Prompting Templates.** Conditioned generation requires rendering structured conditioning variables as natural-language instructions that LLMs can reliably interpret. For reproducibility and fair comparison we provide the full templates used in all experiments in Appendix H, and additionally verify robustness to an alternative template set in Section 5.4.

Table 1: Performance comparison on `ELIX`, `Review`, and `Roleplay` using Avg. Len., Acc., and Win Rate. † indicates a **special token** that is mapped to a learned user-specific latent embedding, rather than a regular token from the model's original tokenizer vocabulary.

| Method | ELIX | | | Review | | | Roleplay | | |
|---|---|---|---|---|---|---|---|---|---|
| | Avg. Len. ↓ | Acc. (%) | Win Rate (%) | Avg. Len. ↓ | Acc. (%) | Win Rate (%) | Avg. Len. ↓ | Acc. (%) | Win Rate (%) |
| Base-Model | – | 50.00 | 50.00 | – | 50.00 | 50.00 | – | 50.00 | 50.00 |
| Raw-Prompting | 3175.08 | 56.12 | 49.31 | 3049.10 | 51.08 | 53.46 | 3020.61 | 53.06 | 35.26 |
| Summary-Prompting | 536.41 | 55.61 | 47.40 | 535.44 | 51.91 | 57.65 | 529.47 | 53.28 | 39.39 |
| POPI-Inference-Only | **52.52** | **71.48** | **63.84** | **320.80** | **91.81** | **80.99** | **200.22** | **54.81** | **55.36** |
| Raw-Aligned | 3175.08 | 73.94 | 56.14 | 3049.10 | 95.07 | **89.54** | 3020.61 | 65.31 | 38.66 |
| Summary-Aligned | 536.41 | 61.57 | 40.41 | 535.44 | 77.64 | 65.24 | 529.47 | 66.14 | 53.45 |
| LoRe | 1.00 † | 63.27 | 51.03 | 1.00 † | 83.54 | 67.76 | 1.00 † | 66.77 | 65.29 |
| PAL-A | 1.00 † | 61.20 | 51.28 | 1.00 † | 91.22 | 78.69 | 1.00 † | 66.60 | 28.41 |
| PAL-B | 1.00 † | 64.80 | 51.68 | 1.00 † | 92.54 | 78.44 | 1.00 † | 66.94 | 33.04 |
| POPI-Full | **52.52** | **80.14** | **63.97** | **320.80** | **95.76** | 88.08 | **200.22** | **72.36** | **70.12** |

Table 2: Generator-transferability on `ELIX`: We compare Win Rate across different personalization methods, when applied on a range of frozen, off-the-shelf LLMs as generators. For each generator, Win Rate (%) is computed relative to that model's own Base-Model.

| Method | Win Rate (%) on ELIX | | | | | | | | Avg. |
|---|---|---|---|---|---|---|---|---|---|
| | Mistral-S | Mistral-L | DeepSeek-R1 | Llama-1b | Llama-11b | Llama-90b | Claude-4 | GPT-4o-mini | |
| Base-Model | 50.00 | 50.00 | 50.00 | 50.00 | 50.00 | 50.00 | 50.00 | 50.00 | 50.00 |
| Raw-Prompting | 67.27 | 72.16 | 79.38 | 27.16 | 56.05 | 59.03 | 72.38 | 71.43 | 63.11 |
| Summary-Prompting | 61.13 | 67.15 | 74.53 | 18.75 | 48.64 | 48.33 | 59.96 | 67.85 | 55.79 |
| Summary-Prompting (Prompt-Dependent Summary) | 55.73 | 51.91 | 66.65 | 43.66 | 41.47 | 44.37 | 58.00 | 57.38 | 52.40 |
| Summary-Prompting (Llama-90b as Inference Model) | 68.36 | 75.09 | 73.51 | 47.25 | 55.60 | 55.15 | 69.39 | 67.50 | 63.98 |
| POPI-Inference-Only | **81.23** | **81.50** | 70.88 | **62.65** | **63.53** | 59.62 | 70.22 | 76.53 | 70.77 |
| POPI-Inference-Only (Prompt-Dependent Summary) | 79.50 | 76.70 | **82.01** | 61.55 | 61.89 | **63.08** | **76.29** | 75.84 | **72.11** |
| POPI-Inference-Only (IPO Based) | 78.05 | 80.57 | 71.75 | 59.78 | 62.79 | 60.99 | 68.23 | **80.83** | 70.37 |

Table 3: Generator-transferability on `Review`: We compare Win Rate across different personalization methods, when applied on a range of frozen, off-the-shelf LLMs as generators. For each generator, Win Rate (%) is computed relative to that model's own Base-Model.

| Method | Win Rate (%) on Review | | | | | | | | Avg. |
|---|---|---|---|---|---|---|---|---|---|
| | Mistral-S | Mistral-L | DeepSeek-R1 | Llama-1b | Llama-11b | Llama-90b | Claude-4 | GPT-4o-mini | |
| Base-Model | 50.00 | 50.00 | 50.00 | 50.00 | 50.00 | 50.00 | 50.00 | 50.00 | 50.00 |
| Raw-Prompting | 48.66 | 58.74 | 79.78 | 34.82 | 67.87 | 70.88 | 70.80 | 71.84 | 62.92 |
| Summary-Prompting | 47.70 | 52.80 | 61.88 | 40.32 | 56.66 | 58.10 | 54.70 | 61.94 | 54.26 |
| POPI-Inference-Only | **72.51** | **85.67** | **84.90** | **77.28** | **83.53** | **79.81** | **74.08** | **81.27** | **79.88** |

Table 4: Generator-transferability on `Roleplay`: We compare Win Rate across different personalization methods, when applied on a range of frozen, off-the-shelf LLMs as generators. For each generator, Win Rate (%) is computed relative to that model's own Base-Model.

| Method | Win Rate (%) on Roleplay | | | | | | | | Avg. |
|---|---|---|---|---|---|---|---|---|---|
| | Mistral-S | Mistral-L | DeepSeek-R1 | Llama-1b | Llama-11b | Llama-90b | Claude-4 | GPT-4o-mini | |
| Base-Model | 50.00 | 50.00 | 50.00 | 50.00 | **50.00** | 50.00 | **50.00** | 50.00 | 50.00 |
| Raw-Prompting | 66.78 | **70.34** | 47.74 | 23.74 | 34.75 | 34.62 | 13.99 | 52.19 | 43.02 |
| Summary-Prompting | **68.62** | 66.31 | 58.18 | 22.41 | 31.25 | 39.85 | 40.65 | 53.49 | 47.60 |
| POPI-Inference-Only | 47.89 | 57.52 | **62.93** | **56.39** | 48.17 | **51.27** | 37.57 | **59.12** | **52.61** |

## 5.2 Results on `ELIX`, `Review`, and `Roleplay`

We evaluate POPI on `ELIX`, `Review`, and `Roleplay`, where ground-truth personas are used *only* at evaluation time for LLM-as-a-judge comparisons and are never observed during personalized generation.

**Personalization Performance and Efficiency.** Table 1 reports main results in two regimes. Without generator fine-tuning, POPI-Inference-Only outperforms Raw-Prompting and Summary-Prompting on all three datasets, with the largest gains on `Review` (win rate 80.99 vs. 57.65) and `ELIX` (63.84 vs. 47.40), and a smaller but consistent improvement on `Roleplay` (55.36 vs. 39.39)—all at roughly an order-of-magnitude reduction in context overhead (e.g., ~53 vs. ~3,175 tokens on `ELIX`). With generator fine-tuning, POPI-Full leads on `ELIX` and `Roleplay`; on `Review`, Raw-Aligned achieves a marginally higher win rate (89.54 vs. 88.08) at ~10× the context cost. LoRe and PAL-A/B minimize context overhead via learned per-user embeddings, but couple representations to a specific generator, precluding the transferability evaluated below.

**Analysis of Learned Summaries.** Figure 3 compares summary lengths before and after optimization on `ELIX`: the unoptimized reference model produces summaries spanning roughly 400–700 tokens, while the optimized model concentrates sharply around 50 tokens, confirming the emergent compression discussed in Section 4.1. We also verify that the optimized summaries remain linguistically well-formed rather than degenerating into adversarial triggers: **99.8% of tokens** are covered by the top 50,000 most frequent English words in the SUBTLEX-US frequency corpus (compared with 98.8% for the Brown Corpus), with no tokens with $\geq 3$ repeated characters (e.g., `aaa`), no symbol-heavy tokens (e.g., `a&&`), no consonant-only tokens of length $\geq 3$ (e.g., `qwty`), and no tokens exceeding 30 characters.

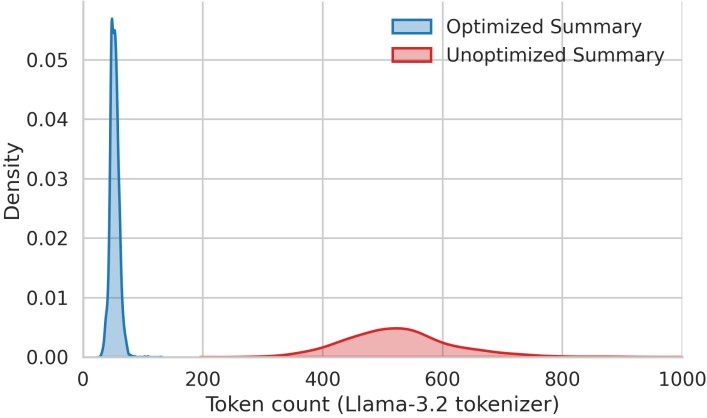

Figure 3: Distribution of preference summary lengths (in tokens) on `ELIX`, before and after optimization. The optimized inference model produces substantially shorter summaries, concentrating around ~50 tokens compared to ~500 for the unoptimized reference model.

**Qualitative Case Study.** Figures 5 and 6 (Appendices F and G) show representative examples on `ELIX`. Before optimization, summaries are verbose (350–467 tokens) and structurally generic across personas; after optimization, they are compact (~50 tokens) and differentiated—for instance, the learned summary includes "child-centric language and simple definitions" for a held-out user with the child ground-truth persona, versus "technical accuracy and technical jargon" for another held-out user with the expert ground-truth persona. These summaries guide generation toward responses aligned with users' preferences.

**Generator-Transferability.** Tables 2–4 evaluate whether optimized summaries remain useful when paired with frozen, off-the-shelf LLMs. On `Review`, POPI-Inference-Only outperforms all baselines on every tested generator, averaging 79.88% win rate versus 62.92% (Raw-Prompting) and 54.26% (Summary-Prompting). On `ELIX`, the pattern is similar (70.77% vs. 63.11% and 55.79%). Notably, we observed that a 3B-parameter inference model optimized with POPI could outperform an unoptimized 90B-parameter model. Raw-Prompting sometimes underperforms despite access to more information, suggesting that unfiltered signals introduce noise that outweighs their informational content. On `Roleplay`, results are more mixed: POPI-Inference-Only averages 52.61% win rate—a modest improvement over the 50% Base-Model baseline and above the below-50% averages of Raw-Prompting (43.02%) and Summary-Prompting (47.60%), but with variance across generators. We attribute this to the low signal-to-noise ratio in Roleplay's user signals: each user is represented by only eight preference pairs spanning diverse topics, with limited stylistic or topical overlap

Table 5: Performance Comparisons on `AlignX` using Avg. Len., Acc., and Win Rate, across four official test splits (Arbitrary, DEMO, PAIR, UGC). We experiment with two different types of user signals: the original heterogeneous user signals (left) and the 90-dimensional preference direction extracted by `AlignX` authors (right).

| Method | AlignX (with Original Heterogeneous User signals) | | | | | | | | AlignX (with 90-Dimensional Preference Directions as User Signals) | | | | | | | |
| --- | --- | --- | --- | --- | --- | --- | --- | --- | --- | --- | --- | --- | --- | --- | --- | --- |
| | Arbitrary | | DEMO | | PAIR | | UGC | | Arbitrary | | DEMO | | PAIR | | UGC | |
| | Avg. Len. | Acc. | Avg. Len. | Acc. | Avg. Len. | Acc. | Avg. Len. | Acc. | Avg. Len. | Acc. | Avg. Len. | Acc. | Avg. Len. | Acc. | Avg. Len. | Acc. |
| Base-Model | – | 50.00 | – | 50.00 | – | 50.00 | – | 50.00 | – | 50.00 | – | 50.00 | – | 50.00 | – | 50.00 |
| Raw-Prompting | 1402.86 | 51.89 | **105.09** | 58.27 | 1501.26 | 49.33 | 1048.37 | 49.06 | 259.92 | 51.97 | 117.94 | 54.90 | 233.88 | 49.81 | 184.68 | 51.19 |
| Summary-Prompting | 288.75 | 52.10 | 138.35 | 57.73 | 315.74 | 48.63 | 284.24 | 49.60 | 257.52 | 52.69 | 186.25 | 55.79 | 248.22 | 50.62 | 239.68 | 50.94 |
| POPI-Inference-Only | **151.65** | **54.63** | 188.25 | **60.37** | **128.04** | **50.22** | **134.94** | **50.92** | **91.47** | **56.55** | **72.83** | **62.72** | **86.30** | **52.80** | **87.97** | **52.67** |
| Raw-Aligned | 1402.86 | **74.60** | **105.09** | **92.51** | 1501.26 | **55.95** | 1048.37 | **57.36** | 259.92 | **74.41** | 117.94 | **87.88** | 233.88 | **59.78** | 184.68 | **58.76** |
| Summary-Aligned | 288.75 | 69.56 | 138.35 | 89.79 | 315.74 | 51.70 | 284.24 | 51.99 | 257.52 | 70.56 | 186.25 | 84.24 | 248.22 | 58.86 | 239.68 | 58.54 |
| POPI-Full | **151.65** | 71.12 | 188.25 | 90.44 | **128.04** | 52.51 | **134.94** | 54.45 | **91.47** | 69.45 | **72.83** | 84.19 | **86.30** | 58.73 | **87.97** | 58.54 |

with the downstream generation. Prepending these user signals directly introduces more noise than useful guidance, actively perturbing downstream generation rather than improving it. POPI's optimization-driven compression mitigates this by partially filtering out non-predictive content, though the performance gains remain modest when the underlying signals carry limited extractable preference information.

## 5.3 Results on `AlignX` Dataset

We next evaluate POPI on `AlignX`, a large-scale forum discussion benchmark that provides two types of user signals (Section 5.1). Table 5 summarizes results across both signal types and four official test splits. Without generator fine-tuning, POPI-Inference-Only outperforms Raw-Prompting and Summary-Prompting on every split and signal type, while generally using much fewer context tokens. The clearest gains appear on `Arbitrary` and `DEMO`, where user signals carry the most preference-relevant information.

When generator fine-tuning is allowed, POPI-Full generally reduces context overhead by a large margin relative to Raw-Aligned, but trails it in accuracy moderately on most splits. A likely contributing factor is the two-stage training design discussed in Section 4.2: because the inference model is optimized against a frozen base generator in Stage 1, it may compress away signal that the base generator cannot exploit but that a subsequently fine-tuned generator could have leveraged. We regard closing this gap—for example, through iterative co-training of both components—as an important direction for future work.

**POPI is complementary to handcrafted user profile extraction methods.** The 90-dimensional preference direction provided with `AlignX` is itself a carefully engineered representation of user preferences (Li et al., 2025). When used as input to POPI's inference model, the optimized summaries improve accuracy on every split (e.g., 56.55% vs. 51.97% on `Arbitrary`) while compressing context (e.g., from ∼260 to ∼88 tokens on `Arbitrary`). This suggests that POPI's optimization can serve as a refinement layer on top of existing handcrafted user profile extraction pipelines, rather than as a replacement.

## 5.4 Ablation Studies

We conduct four ablation studies on the `ELIX` dataset to better understand the design choices of POPI: **(1)** the effect of using IPO as the underlying framework, **(2)** the effect of prompt-dependent summaries, **(3)** sensitivity to the scaling parameter $\beta$ (Appendix D), and **(4)** robustness to prompting templates (Appendix E).

**IPO as the Underlying Framework.** In this study, we adopt IPO as the underlying preference alignment framework for POPI and baselines, instead of DPO, as described in Section 4.3. Main results are shown in Table 6, with transfer results reported in Table 2. Across both settings, POPI based on IPO performs comparably to DPO across all metrics, confirming that POPI is broadly compatible with different underlying alignment frameworks.

**Prompt-Dependent Summaries.** Table 7 reports results for the prompt-dependent variant described in Section 4.4, where the summary additionally conditions on the current prompt $x$. Conditioning on the prompt yields modest improvements over prompt-independent summaries, suggesting that most personalization signal

Table 6: Performance comparisons on `ELIX` using Avg. Len., Acc., and Win Rate, with the summary-augmented IPO objective.

| Method | ELIX (IPO Based) | | |
|---|---|---|---|
| | Avg. Len. ↓ | Acc. (%) | Win Rate (%) |
| Base-Model | – | 50.00 | 50.00 |
| Raw-Prompting | 3175.08 | 56.12 | 49.31 |
| Summary-Prompting | 536.41 | 55.61 | 47.40 |
| POPI-Inference-Only | **60.24** | **69.07** | **63.78** |
| Raw-Aligned | 3175.08 | 68.88 | 55.80 |
| Summary-Aligned | 536.41 | 62.77 | 45.09 |
| POPI-Full | **60.24** | **79.98** | **64.00** |

is captured by user-level summaries derived from $s_i$ alone. This empirically supports our user-level scoping assumption: stable, prompt-independent preferences account for the majority of the personalization benefit.

Table 7: Performance comparisons on `ELIX` with the prompt-dependent preference summary. We report average additional input length (Avg. Len.), accuracy (Acc.), and Win Rate.

| Method | ELIX (Prompt-Dependent Summary) | | |
|---|---|---|---|
| | Avg. Len. ↓ | Acc. (%) | Win Rate (%) |
| Base-Model | – | 50.00 | 50.00 |
| Raw-Prompting | 3175.08 | 56.12 | 49.31 |
| Summary-Prompting | 543.66 | 54.18 | 43.32 |
| POPI-Inference-Only | **111.38** | **71.11** | **66.36** |
| Raw-Aligned | 3175.08 | 73.94 | 56.14 |
| Summary-Aligned | 543.66 | 60.20 | 43.06 |
| POPI-Full | **111.38** | **81.65** | **65.37** |

## 6 Conclusion

We introduced POPI, a framework for user-level LLM personalization that separates the problem into preference inference and conditioned generation connected by a natural-language interface. A shared inference model, trained via reinforcement learning against a summary-augmented preference-optimization objective, distills heterogeneous user signals into concise natural-language preference summaries; a shared generator then conditions on these summaries to produce personalized responses. Our experiments across four benchmarks suggest that directly optimizing the preference-inference step—rather than treating it as heuristic preprocessing—is an effective lever for personalization: it produces compact summaries that reduce context overhead by a large margin while generally improving alignment with user preferences across the settings we tested. Because the interface is natural language, the learned summaries can be reused with frozen, off-the-shelf generators—including black-box commercial APIs—without retraining, and the optimization is complementary to existing handcrafted preference-extraction pipelines, providing further gains when applied as a refinement layer.

**Limitations and Future Work.** The two-stage training design reflects a trade-off: optimizing the inference model against a frozen base generator encourages transferable summaries, but may compress away signal that a jointly fine-tuned generator could exploit. Iterative but constrained co-training that preserves transferability while retaining richer signal is a natural next step. More broadly, when user signals are especially sparse or heterogeneous, the extractable preference information may itself be limited; understanding when and why preference inference is structurally hindered under such conditions remains an open question. Finally, our framework assumes relatively stable user preferences; extending it to preferences that evolve over time would expand its applicability and connect to emerging work on continual adaptation in language models.

**Broader Impact Statement**

This paper aims to advance the scientific understanding of user-level personalization in large language models through a modular framework for preference inference and conditioned generation. Potential societal impacts largely align with those typical of research on machine learning personalization: improved user experience, more adaptable model outputs, and broader accessibility for individuals with diverse needs. As with any personalization method, practical deployments should consider standard concerns such as privacy, transparency, and responsible use of user-provided signals. Beyond these well-established considerations, we do not identify additional ethical or societal risks unique to the methods proposed in this work.

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

## Appendix

## A   Training and Inference Algorithms

Algorithms 1 and 2 summarize the two-stage training procedure and the inference procedure of POPI, respectively, as described in Sections 4.1 and 4.2.

---
**Algorithm 1** Training Procedure of POPI

---
**input** User signals $\{s_i\}_{i=1}^N$, preference-labeled datasets $\{\mathcal{D}_i\}_{i=1}^N$, reference inference model $\pi_{\phi_{\mathrm{ref}}}$, reference generator $\pi_{\theta_{\mathrm{ref}}}$

**output** Trained inference model $\pi_\phi$, (optional) trained generator $\pi_\theta$

 1: Initialize $\pi_\phi \leftarrow \pi_{\phi_{\mathrm{ref}}}$
 2: Initialize $\pi_\theta \leftarrow \pi_{\theta_{\mathrm{ref}}}$
 3: **Stage 1: Optimizing $\pi_\phi$**
 4: **for** each training iteration **do**
 5:    Sample user $i$ and example $(x, y_c, y_r) \sim \mathcal{D}_i$
 6:    Sample a group of preference summaries $z_i \sim \pi_\phi(\cdot \mid s_i)$
 7:    **for** each sampled $z_i$ **do**
 8:       Compute the reward as the negative summary-augmented objective $-\ell_i$ using Equation 1
 9:    **end for**
10:    Update $\pi_\phi$ with **GRPO**, with regularization $\alpha\,\mathrm{KL}(\pi_\phi(\cdot \mid s_i)\,\|\,\pi_{\phi_{\mathrm{ref}}}(\cdot \mid s_i))$
11: **end for**

12: **Stage 2 (Optional): Optimizing $\pi_\theta$**
13: **for** each training iteration **do**
14:    Sample user $i$ and example $(x, y_c, y_r) \sim \mathcal{D}_i$
15:    Sample preference summary $z_i \sim \pi_\phi(\cdot \mid s_i)$
16:    Compute the summary-augmented objective $\ell_i$ using Equation 1
17:    Update $\pi_\theta$ by minimizing $\ell_i$ via standard backpropagation
18: **end for**

---

## B   Derivation of the Decomposition in Equation 3

We derive the information-theoretic decomposition of the summary-augmented DPO objective $\mathcal{L}$. Following the standard treatment in Rafailov et al. (2023), we begin by deriving an implicit reward model parameterized

---

**Algorithm 2** Inference Procedure of POPI

---

**input** User signals $s_i$, user prompt(s) $\{x\}$, trained inference model $\pi_\phi$, trained, frozen, or black-box generator $\pi_\theta$

**output** Personalized response(s) $\{y\}$

1: Generate a preference summary $z_i \sim \pi_\phi(\cdot \mid s_i)$
2: **for** each user prompt $x$ **do**
3:     Generate a personalized response $y \sim \pi_\theta(\cdot \mid x, z_i)$
4: **end for**
5: **return** $\{y\}$

---

by the generation model $\pi_\theta$:

$$
r_{\pi_\theta}(x, y, z_i) = \beta \log \frac{\pi_\theta(y \mid x, z_i)}{\pi_{\theta_{\mathrm{ref}}}(y \mid x)} + \beta \log Z(x, z_i),
$$

$$
Z(x, z_i) = \sum_y \pi_{\theta_{\mathrm{ref}}}(y \mid x) \exp(\frac{1}{\beta} r_{\pi_\theta}(x, y, z_i)). \tag{4}
$$

Under the Bradley–Terry (BT) model of preferences, the conditional probability that $y_c$ is preferred over $y_r$, given prompt $x$ and summary $z_i$, is expressed as:

$$
P_{\pi_\theta}(y_c \succ y_r \mid x, z_i) = \sigma\Big(r_{\pi_\theta}(x, y_c, z_i) - r_{\pi_\theta}(x, y_r, z_i)\Big). \tag{5}
$$

Substituting this into the $\mathcal{L}$ formulation, we can write it as:

$$
\mathcal{L} = \frac{1}{N} \sum_{i=1}^{N} \mathbb{E}_{(x, y_c, y_r) \sim \mathcal{D}_i, \; z_i \sim \pi_\phi(\cdot \mid s_i)} \Big[ -\log P_{\pi_\theta}(y_c \succ y_r \mid x, z_i) \Big]. \tag{6}
$$

We then decompose this objective to reveal its information-theoretic structure:

$$
\begin{aligned}
\mathcal{L} = \frac{1}{N} \sum_{i=1}^{N} \mathbb{E}_{(x, y_c, y_r), z_i} & \left[ \log \frac{P(y_c \succ y_r \mid x, z_i)}{P_{\pi_\theta}(y_c \succ y_r \mid x, z_i)} \right. \\
& \left. + \log \frac{1}{P(y_c \succ y_r \mid x)} - \log \frac{P(y_c \succ y_r \mid x, z_i)}{P(y_c \succ y_r \mid x)} \right] \\
= \; & \mathrm{KL}(P(y_c \succ y_r \mid x, z_i) \,\|\, P_{\pi_\theta}(y_c \succ y_r \mid x, z_i)) \\
& + H(P(y_c \succ y_r \mid x)) - I(y_c \succ y_r; z_i \mid x),
\end{aligned} \tag{7}
$$

which is the decomposition stated in Equation 3. The entropy term $H(P(y_c \succ y_r \mid x))$ is independent of both models' parameters and therefore acts as a constant throughout training, so optimization of $\mathcal{L}$ reduces to jointly minimizing the KL term (generator approximation error) and maximizing the conditional mutual information (summary informativeness), as discussed in the main text.

## C  Stability of RL Training

We analyze the stability of RL by tracking the mean and standard deviation of the reward during training on the `Review` dataset. As shown in Figure 4, the average reward increases steadily, while the variance decreases and stabilizes over time. This behavior indicates stable and well-behaved optimization of the preference inference model under our training setup.

## D  Ablation: Effect of the Scaling Parameter

Table 8 examines sensitivity to the hyperparameter $\beta \in \{0.1, 0.05, 0.01\}$, which scales the implicit reward model. POPI performs consistently across this range, with $\beta = 0.01$ yielding the best results.

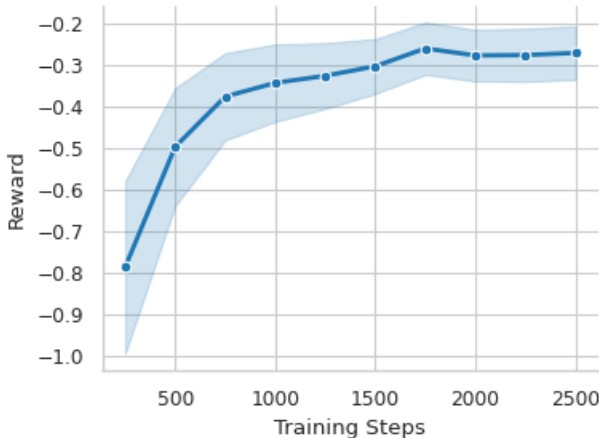

Figure 4: Mean reward (solid line) and standard deviation (shaded area) during GRPO training on the `Review` dataset.

Table 8: Ablation study on the hyperparameter $\beta$ on `ELIX` using Avg. Len., Acc., and Win Rate.

| Method | $\beta$ | ELIX | | |
| --- | --- | --- | --- | --- |
| | | Avg. Len. $\downarrow$ | Acc. (%) | Win Rate (%) |
| Base-Model | – | 0.00 | 50.00 | 50.00 |
| POPI-Inference-Only | 0.10 | 118.80 | 68.43 | 59.92 |
| | 0.05 | 145.92 | 70.42 | 62.97 |
| | 0.01 | **52.52** | **71.48** | **63.84** |
| POPI-Full | 0.10 | 118.80 | 76.54 | 53.91 |
| | 0.05 | 145.92 | 79.69 | 59.05 |
| | 0.01 | **52.52** | **80.14** | **63.97** |

## E  Ablation: Robustness to Prompting Templates

To assess whether our results depend on a particular prompting template, we re-ran the generator-transferability experiments on `ELIX` using a different set of templates (Figure 8). Table 9 reports the results: although individual models exhibit some variance across templates, the averaged performance remains consistent, indicating that the gains from optimized preference summaries are not driven by template-specific effects.

Table 9: Comparison of win rates on `ELIX` using an alternative prompting template set, across off-the-shelf LLMs. The win rates are computed relative to each LLM's own Base-Model. Cells are color-coded, where darker blue indicates lower values, and lighter red indicates higher values. Numbers in parentheses indicate changes relative to the original templates.

| Method | Win Rate (%) on ELIX | | | | | | | | Avg. |
| --- | --- | --- | --- | --- | --- | --- | --- | --- | --- |
| | Mistral-S | Mistral-L | DeepSeek-R1 | Llama-1b | Llama-11b | Llama-90b | Claude-4 | GPT-4o-mini | |
| Base-Model | 50.00(+0.00) | 50.00(+0.00) | 50.00(+0.00) | 50.00(+0.00) | 50.00(+0.00) | 50.00(+0.00) | 50.00(+0.00) | 50.00(+0.00) | 50.00(+0.00) |
| Raw-Prompting | 58.23(-9.04) | 74.00(+1.84) | **77.28(-2.10)** | 31.38(+4.22) | 58.70(+2.65) | **57.98(-1.05)** | **75.47(+3.09)** | 69.57(-1.86) | 62.83(-0.28) |
| Summary-Prompting | 56.17(-4.96) | 65.50(-1.65) | 69.22(-5.31) | 19.02(+0.27) | 49.06(+0.42) | 50.00(+1.67) | 59.74(-0.22) | 65.65(-2.20) | 54.30(-1.50) |
| POPI-Inference-Only | **75.98(-5.25)** | **83.95(+2.45)** | 70.32(-0.56) | **67.93(+5.28)** | **66.10(+2.57)** | 57.88(-1.74) | 70.63(+0.41) | **82.76(+6.23)** | **71.94(+1.17)** |

# F   Qualitative Illustration: Summary Before vs. After Optimization

| Ground Truth Persona | Unoptimized Preference Summary (Truncated for Brevity) | Optimized Preference Summary |
|---|---|---|
| The user is a child in elementary school. | ## User Preference Analysis and Summary

In analyzing the provided examples, common patterns and preferences emerge in user responses to the same prompts. These patterns can be summarized as follows:

### 1. **Tone and Language**

* **Preferred Response**: Uses formal and structured language, providing clear explanations and definitions.
* **Dispreferred Response**: Utilizes informal language, often with a conversational tone and fragmented sentences.

### 2. **Depth of Information**
*[... 403 tokens omitted ...]* | **Tone:**
Preferred responses:
- Use child-centric language and simple definitions
- Basic examples and analogies
- Young child's perspective
- Young child's grammar and sentence structure
- Informal, conversational tone |
| The user is a preteen in middle school. | # User Preference Patterns

## Overview

Upon analyzing the provided text examples, several patterns emerge in the user's preferred and dispreferred responses. The following summary captures the most notable differences and generalizes them to guide the generation of personalized responses.

### Key Patterns

1. **Direct and Concise Language**
     * Preferred responses tend to be direct, clear, and to the point.
*[... 350 tokens omitted ...]* | **Tone and Language:**
Preferred responses:
- Use child-centric language and simple definitions
- Basic examples and analogies
- Informal, conversational tone
- Basic technical accuracy, but may lack technical jargon or technical accuracy
- Informal and casual |
| The user is a teenager in high school. | **User Preference Patterns in Response Generation**

The examples provided showcase distinct approaches to generating responses to user prompts. Analyzing these patterns can help identify user preferences and inform strategies for creating personalized responses.

**Key Findings:**

1. **Depth of Explanation:** Preferred responses tend to provide more detailed explanations, often incorporating supporting evidence and scientific concepts. Dispreferred responses provide shorter, more general explanations that focus on the main idea.
2. **Organization:** Preferred responses often follow a structured format, breaking down complex topics into manageable sections or steps.
*[... 392 tokens omitted ...]* | **Tone:**
Preferred responses:
- Use conversational language and simple definitions
- Basic examples and analogies
- Informal, casual tone
- Basic technical accuracy, but may not use technical jargon
- Show willingness to help and explain concepts
- Examples and real-life scenarios |
| The user is a young adult in college. | Based on the analysis of the four examples, I have identified common patterns and user preferences in their responses. Here is a concise and structured summary that captures these preferences:

**User Preferences:**

1. **Clear and concise explanations**: Users prefer brief, straightforward explanations that get straight to the point, avoiding unnecessary complexity or jargon.
2. **Detailed information**: Users expect detailed information on the topic, especially when it comes to technical or scientific concepts. They want to understand the underlying principles and mechanisms.
3. **Organization and structure**: Users appreciate well-organized responses that break down complex information into logical sections or subheadings. This makes it easier to follow and understand the content.
*[... 467 tokens omitted ...]* | **Tone:**
Preferred responses:
- Use technical accuracy and technical jargon
- Professional tone, but approachable and conversational
- Provide additional information and examples to clarify concepts
- Acknowledge the user's expertise and offer to help with follow-up questions
- Use proper grammar and sentence structure |
| The user is an expert. | **User Preference Analysis and Summary**

**Key Findings:**

1. **Structure and Format:** Preferred responses tend to follow a more formal and structured format, often starting with a clear topic introduction, followed by a step-by-step explanation, and concluding with a summary or final thoughts. Dispreferred responses, on the other hand, are often more conversational and less structured.
2. **Tone and Language:** Preferred responses tend to use more technical vocabulary and jargon, while dispreferred responses use simpler language and avoid technical terms. This suggests that users prefer more detailed and specialized explanations when they are familiar with the topic.
3. **Depth of Information:** Preferred responses generally provide more in-depth information, including examples and supporting details, whereas dispreferred responses tend to focus on a more general overview.
*[... 438 tokens omitted ...]* | ***Tone:**
Preferred responses:
- Use technical accuracy and technical jargon
- Professional tone
- Provide additional information and examples
- Acknowledge user expertise and offer to elaborate on the topic
- Use proper grammar and sentence structure |

Figure 5: Comparison of preference summaries before and after optimization on the `ELIX` dataset (same thermostat example as Figure 6). Unoptimized summaries are truncated for space; token counts are annotated. The unoptimized inference model produces verbose, structurally generic summaries across all personas, while the optimized model produces compact, persona-specific summaries that capture distinct preference profiles.

# G   Qualitative Illustration: Summary and Personalized Generation

| Ground Truth Persona | Optimized Preference Summary | Personalized Response (Truncated for Brevity) |
|---|---|---|
| The user is a child in elementary school. | **Tone:** Preferred responses: - Use child-centric language and simple definitions - Basic examples and analogies - Young child's perspective - Young child's grammar and sentence structure - Informal, conversational tone | Temperature is like how hot or cold it is in our house, right? So, there's this thing called a thermostat. It's like a special helper that keeps our house at a comfy temperature. It controls the heat or the cold air.

There's a tiny sensor inside the thermostat that feels the temperature in our house. This sensor tells the thermostat two things: how hot or cold it is, and how hot or cold it should be. When it gets too hot or too cold, it sends a message to a big machine called a heater or a cooler, depending on what we need.

When the thermostat gets the message that it's too hot, it sends the heater to turn off the heat. And when it's too cold, it sends the heater to turn on the heat. It's like a seesaw, it helps balance the temperature so our house doesn't get too hot or too cold. When the temperature goes back to the right amount, the thermostat says "okay" and the heater or cooler stops working. *[...]* |
| The user is a preteen in middle school. | **Tone and Language:** Preferred responses: - Use child-centric language and simple definitions - Basic examples and analogies - Informal, conversational tone - Basic technical accuracy, but may lack technical jargon or technical accuracy - Informal and casual | Let's talk about thermostats. So, a thermostat is a device that helps regulate the temperature in a room or even an entire house. Its main job is to keep the temperature at a comfortable level for you and your family.

Here's how it works:

1. A thermostat has two main parts: a sensor and a heating/cooling unit (like a heating element or a coil). The sensor detects the current temperature in the room, and the unit does the actual work to change the temperature.
2. When the temperature in the room goes above or below a certain set point (like 72 degrees Fahrenheit, for example), the sensor sends a signal to the thermostat.
3. The thermostat then decides whether to turn on or turn off the heating or cooling unit. If the temperature is too high, it turns on the heating unit to warm it back down. If it's too low, it turns on the cooling unit to cool it back up. *[...]* |
| The user is a teenager in high school. | **Tone:** Preferred responses: - Use conversational language and simple definitions - Basic examples and analogies - Informal, casual tone - Basic technical accuracy, but may not use technical jargon - Show willingness to help and explain concepts - Examples and real-life scenarios | A thermostat is a pretty common device in our homes and offices that helps regulate the temperature. So, let's break it down simply.

Here's how it works:

1. A thermostat has two main parts: a sensor and a heating/cooling element. The sensor detects the current temperature in the space, like the room you're trying to heat or cool.
2. There's usually an electronic circuit that sends the temperature reading to a microchip, which is like a small computer within the thermostat.
3. The microchip compares the current temperature to a set point, or the desired temperature you've programmed into the thermostat (usually the temperature you like your space to be in, like 72°F in the winter and 78°F in the summer). *[...]* |
| The user is a young adult in college. | **Tone:** Preferred responses: - Use technical accuracy and technical jargon - Professional tone, but approachable and conversational - Provide additional information and examples to clarify concepts - Acknowledge the user's expertise and offer to help with follow-up questions - Use proper grammar and sentence structure | I'd like to break down how a thermostat works in a way that's easy to understand.

A thermostat is a fundamental device that plays a crucial role in maintaining a consistent temperature in your home or building. Its primary function is to regulate the temperature by sensing the current temperature and adjusting it to a setpoint, which is the desired temperature you've programmed. This process is based on a simple yet reliable principle.

Here's a step-by-step explanation:

1. **Sensing the current temperature**: Most modern thermostats use one of two common methods to measure the current temperature: thermocouples or thermistors. A thermocouple is a device that generates a small voltage based on the temperature difference between two dissimilar metals, while a thermistor is a resistor whose resistance changes in response to temperature changes. These sensors send the temperature data to the thermostat's control circuit. *[...]* |
| The user is an expert. | ***Tone:** Preferred responses: - Use technical accuracy and technical jargon - Professional tone - Provide additional information and examples - Acknowledge user expertise and offer to elaborate on the topic - Use proper grammar and sentence structure | I'd be happy to explain how a thermostat controls temperature in a detailed and technical manner.

A thermostat is a fundamental device that plays a crucial role in maintaining a consistent temperature within a building or a system, regulating the thermal energy to achieve a desired setpoint. At its core, a thermostat functions by utilizing a simple yet effective principle: the concept of thermal feedback control.

There are two primary types of thermostats: analog and digital. Analog thermostats use a bimetallic strip, which expands or contracts in response to changes in temperature, while digital thermostats rely on electronic sensors to measure the temperature and send signals to the control unit.

Here's the process:

1. **Sensing**: The thermostat contains a temperature sensor, typically a thermistor (a type of resistor whose resistance changes with temperature) or a thermocouple, which detects the current temperature of the environment it's monitoring. This sensor provides an analog signal that reflects the actual temperature. *[...]* |

Figure 6: Qualitative case study on the ELIX dataset. Each row corresponds to one of the five ground-truth personas in ELIX, all responding to the same user prompt: "How does a thermostat control temperature?" The ground-truth personas are not accessible during training. For each persona, we show an example of the optimized preference summary inferred from user signals, and the corresponding personalized response generated (truncated for brevity). The summaries capture stylistic and content-level preferences at different expertise levels, while the generated responses demonstrate how conditioning on these summaries yields appropriately tailored explanations.

# H   Preference Inference and Personalized Generation Prompting Templates

| System- and user-level prompting template for the Preference Inference LLM, outputting a preference summary | | | | System- and user-level prompting template for the Generation LLM, outputting a (personalized) response | | | |
|---|---|---|---|---|---|---|---|
| **Conditions** | **Dataset** | **System Prompt** | **User Prompt Template** | **Conditions** | **Dataset** | **System Prompt** | **User Prompt Template** |
| User Signals | AlignX (with original user signals) | You are a helpful assistant. | # User Information and Historical Behavior {user_signals}

# Instruction Write a concise and structured summary of the User Information and Historical Behavior above. Your summary should be suitable for guiding other Large Language Models to write Reddit-style comments that mimic this user. | User Prompt | AlignX (both) | You are writing a Reddit-style comment in first person. | # Instruction Write a Reddit-style comment on the post below.

# Post {user_prompt} |
| | AlignX (with extracted preference direction) | | # User Traits {user_signals}

#Instruction Convert the above User Traits into a usable and concise user summary. This summary should be suitable for guiding other Large Language Models to write Reddit-style comments that mimic this user. | | ELIX, Review, Roleplay | You are a helpful assistant. | {user_prompt} |
| | | | # User Preference Examples {user_signals}

# Instruction You are an expert at identifying and summarizing user preferences from texts. Analyze the examples above, where each consists of a preferred and dispreferred response to the same prompt. Based on these comparisons, write a concise and structured summary that captures the user's preferences. Your summary should generalize beyond the specific examples and be suitable for guiding the generation of future personalized responses | User Signals, User Prompt | AlignX (with original user signals) | You are writing a Reddit-style comment in first person that mimics a given user. | # User Information and Historical Behavior {user_signals}

# Instruction Write a Reddit-style comment on the post below, mimicking the user described above.

# Post {user_prompt} |
| | | | | | AlignX (with extracted preference direction) | | # User Traits {user_signals}

# Instruction Write a Reddit-style comment on the post below, mimicking the user described above.

# Post {user_prompt} |
| User Signals, User Prompt | ELIX, Review, Roleplay | | # User Preference Examples {user_signals}

# Instruction You are an expert at identifying and summarizing user preferences from texts. Analyze the examples above, where each consists of a preferred and dispreferred response to the same prompt. Based on these comparisons, write a concise and structured summary that captures the user's preferences. Your summary will be used to guide the generation of personalized responses for the user's upcoming prompt.

# User Upcoming Prompt {user_prompt} | | ELIX, Review, Roleplay | You are a helpful assistant. | # User Preference Examples {user_signals}

# Instruction Given the examples above, generate a preferred response to the user prompt below.

# User Prompt {user_prompt} |
| | | | | Preference Summary, User Prompt | AlignX (both) | You are writing a Reddit-style comment in first person that mimics a given user. | # User Summary {preference_summary}

# Instruction Write a Reddit-style comment on the post below, mimicking the user described above.

# Post {user_prompt} |
| | | | | | ELIX, Review, Roleplay | You are a helpful assistant. | # User Preference Summary {preference_summary}

# Instruction Given the user preference summary above, generate a personalized response to the user prompt below.

# User Prompt {user_prompt} |

Figure 7: Left: prompting templates for the preference inference LLM, which transforms raw user signals (or user signals plus prompt) into natural language summaries. Right: prompting templates for the generation LLM, which integrate user prompts, raw signals, or preference summaries to produce personalized responses. These templates render conditioning variables as natural language.

# I  Alternative Personalized Generation Prompting Templates for Table 9

| Alternative system- and user-level prompting template for the Generation LLM, outputting a (personalized) response | | | |
|---|---|---|---|
| **Conditions** | **Dataset** | **System Prompt** | **User Prompt Template** |
| User Prompt | ELIX, Review, Roleplay | You are a helpful assistant who provides well-considered responses. | [Instruction]
Craft a response to the User Prompt.

[User Prompt]
{user_prompt} |
| User Signals, User Prompt | | You are a helpful assistant who adapts responses according to user preferences. | [Instruction]
Use the User Preference Examples provided below as guidance to craft a personalized response to the User Prompt.

[User Preference Examples]
{user_signals}

[User Prompt]
{user_prompt} |
| Preference Summary, User Prompt | | | [Instruction]
Use the User Preference Summary provided below as guidance to craft a personalized response to the User Prompt.

[User Preference Summary]
{preference_summary}

[User Prompt]
{user_prompt} |

Figure 8: Alternative prompting templates for the generation LLM, used in the robustness ablation study (Section 5.4, Table 9). These templates differ from the primary templates shown in Figure 7 in structure, wording, and length, while preserving the same conditioning variables.

# J  LLM-as-a-Judge Prompting Template

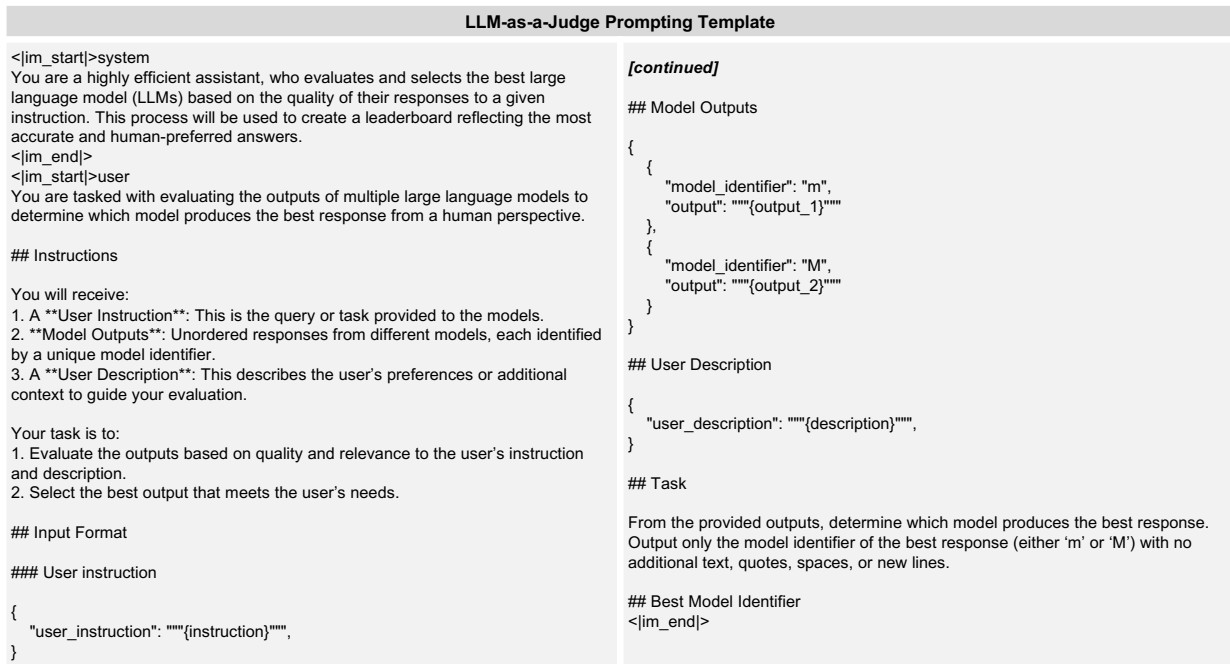

| LLM-as-a-Judge Prompting Template |
|---|
| <\|im_start\|>system
You are a highly efficient assistant, who evaluates and selects the best large language model (LLMs) based on the quality of their responses to a given instruction. This process will be used to create a leaderboard reflecting the most accurate and human-preferred answers.
<\|im_end\|>
<\|im_start\|>user
You are tasked with evaluating the outputs of multiple large language models to determine which model produces the best response from a human perspective.

## Instructions

You will receive:
1. A **User Instruction**: This is the query or task provided to the models.
2. **Model Outputs**: Unordered responses from different models, each identified by a unique model identifier.
3. A **User Description**: This describes the user's preferences or additional context to guide your evaluation.

Your task is to:
1. Evaluate the outputs based on quality and relevance to the user's instruction and description.
2. Select the best output that meets the user's needs.

## Input Format

### User instruction

{
  "user_instruction": """{instruction}""",
} |

```
[continued]

## Model Outputs

{
    {
        "model_identifier": "m",
        "output": """{output_1}"""
    },
    {
        "model_identifier": "M",
        "output": """{output_2}"""
    }
}

## User Description

{
    "user_description": """{description}""",
}

## Task

From the provided outputs, determine which model produces the best response.
Output only the model identifier of the best response (either 'm' or 'M') with no
additional text, quotes, spaces, or new lines.

## Best Model Identifier
<|im_end|>
```

Figure 9: The template used when querying GPT-4o to serve as a judge. The template ensures that the model considers both the user's prompt and the ground-truth user description when comparing candidate outputs.

