# OpenReview forum: "POPI: Personalizing LLMs via Optimized Natural Language Preference Inference"
_TMLR — Under review for TMLR_

### Review · Reviewer_7xvr · 2026-05-22

**Summary Of Contributions:**

The paper presents a general approach for LLM personalization. The technique uses two LLMs, one for generating textual preference summaries based on user information, and another for generating answers to prompts conditioned on preference summaries. The two LLMs can be fine-tuned using algorithms for reinforcement learning from human feedback.


**Pros**

The proposed approach is generic and allows to personalize black-box LLMs.

The experiments are quite comprehensive, which empirically demonstrate the advantages and potential limitations of their propositions. In particular, I appreciate that the authors test various variations of their approach.

The paper is generally well-written and clear.

**Cons**

The proposed method is quite straightforward and basically uses prompt optimization, which I believe is an active research direction. I would suggest the authors to discuss this line of work (in particular, those using human preference) and contrast it with their proposition.

The performance of the two-stage training method lags behind a simple baseline (Raw-Aligned). While I appreciate the authors commenting this issue and providing a conjecture about why it fails. I think it would have been better to demonstrate a method that really works. Otherwise, it somewhat invalidates the proposition of two-stage training.
I would have appreciated more investigation to understand the issue. For instance, would adding a regularization term to control the length of the summary be helpful?

**Additional Comments:**

**Minor points**:

The positioning of figures and tables could be improved. I think it would be better that Section 1 doesn't start with Figure 1 and also, it would be better if figures and tables are closer to the text that refers to them.

"Generator-transferability" makes me think that the generator is transferable. However, in this work, is it the other model that is transferable.

In Fig. 2, should the user signals be denoted s_i instead of c_i?

I would consider DPO also as an RLHF method, since it is how it was derived. Referring to it as a "Standard Back-Propagation and Gradient Descent" is a bit confusing in my opinion.

In addition, referring to RLHF algorithms as "average-user preference alignment algorithms" is a bit of mischaracterization, since they can be used both with one single user and with many users. Nothing in those algorithms require many users, but in practice, it does happen that they are used with preferences from many users.

References not part of sentences should be between parentheses, e.g.,
Page 8: Zhou et al. (2024b) -> (Zhou et al. 2024b)


**Question**:

For the hyperaparameters of DPO, how was 0.002 chosen? Was the grid search for \beta done for each baseline methods?

**Audience:**

Yes

**Audience Explanation:**

Personalization of LLMs is an on-going active research direction. Researchers in this direction will certainly be interested in this work.

**Broader Impact Concerns:**

The existing discussion provided by the authors is sufficient in my opinion.

**Claims And Evidence:**

Yes

**Claims Explanation:**

The authors make two claims:

(1) "POPI generally improves personalization quality while reducing context overhead by up to an order of magnitude."

(2) "the learned summaries transfer to frozen, off-the-shelf generators."

I believe both are demonstrated in their experiments.

**Requested Changes:**

The related work on prompt optimization should be discussed.

I think addressing the two-stage training issue mentioned above in a more consequent way (e.g., enforcing longer summaries, or iterative two-stage training) than discussing it would make the paper stronger, although I understand this would require some effort.

---

### Review · Reviewer_yrLC · 2026-05-23

**Summary Of Contributions:**

This paper proposes a framework for learning personalized LLMs. The framework uses two LLMs: a preference inference LLM, for summarizing the user persona per user, and a conditioned generation LLM, for generating the response per query conditioned on the generated user persona summary. The conditioned generation LLM can be fine-tuned for personalized responses using different alignment methods (e.g., DPO and IPO) with user summary appended in the context, while the preference inference LLM can be trained using RL (e.g., GRPO) with the alignment loss as the reward for the generated user summary. Empirically, the paper mainly demonstrates 1) the preference inference LLM increases performance while compressing the user summary, 2) if the conditioned generation LLM is also fine-tuned, performance can be further improved, and 3) once trained, the preference inference LLM can also sometime bring performance improvements with other downstream conditioned generation LLMs.

The **strengths** of the paper are
- The empirical investigation appears to be comprehensive and of good quality. Empirical results cover multiple datasets, and the utility of the trained, preference inference LLMs is validated with a wide range of downstream LLMs, including commercial ones.
- Overall, the paper demonstrates the usefulness of using LLM to summarize user persona from user-specific data (either raw or processed). The results should be of interest to the LLM community.
- The paper is well-written and easy to follow.

The paper’s main **weaknesses** are
- All results do not include any statistical measure. While the conclusions across different datasets or models are often consistent or correlated, no statistical measure is provided for individual results. Also, the number of data points to calculate the evaluation metric is not provided for all results.
- The two-stage training design is motivated by a trade-off between transferability and fine-tuning gains, but this trade-off is asserted (Section 4.2 and Conclusion) but not demonstrated empirically. Specifically, the paper does show that the inference models trained with frozen generators do transfer, but it does not show fine-tuning gains hurt transferability. It would be interesting to substantiate the claim by conducting further experiments. For example, one can update inference models with updated generators and show that they could exhibit poorer transfer performance.
- Limited analysis of the learned summary. While the paper provides qualitative examples from ELIX, the current scope of qualitative analysis is quite limited. Providing examples across datasets, especially from the Roleplay dataset when the proposed approach does not help (e.g., POPI with Claude-4 in Table 4), would be quite informative and further strengthen the paper.

**Audience:**

Yes

**Audience Explanation:**

Personalization of LLMs is both an interesting research question and a direction for improving industrial LLM systems. The paper provides a useful LLM personalization approach and relevant empirical data points.

**Broader Impact Concerns:**

The current broader impact statement sufficiently addresses relevant concerns.

**Claims And Evidence:**

No

**Claims Explanation:**

As mentioned in the first point of weaknesses above, the empirical results do not include any statistical measure, while the test set sizes are also not reported. Thus, it is unclear how reliable the empirical results are to support the claims in Section 5.

**Requested Changes:**

Changes or questions that are crucial for a recommendation for acceptance:
- Some statistical measure (e.g., bootstrap confidence intervals) of the point estimates reported in the main results (Tables 1-5) should be provided.
- The test set sizes should also be reported for all results (i.e., how many samples are used to generate the table entry).
- For the AlignX experiments, why is Win Rate not reported? Is this a limitation of the evaluation protocol or a deliberate choice?


Other suggestions that can strengthen the paper but are not necessary for acceptance recommendation:
- Adding more analysis to the learned summary, especially for the failure cases, would give valuable insights and strengthen the paper.
- Settling the trade-off between transferability and fine-tuning gains would also strengthen the contributions of the paper.
- On page 5, $\sigma$ is not defined in the DPO objective. While readers familiar with DPO may be able to infer that $\sigma$ represents the logistic function, the paper should provide an explanation or definition of the symbols it introduces.

---

### Review · Reviewer_jsRb · 2026-07-11

**Summary Of Contributions:**

The paper proposes a modular framework for user-level LLM personalization with two components, (i) preference inference map $f:s_i\rightarrow z_i$, (ii) conditional generation $g(x, z_i)\rightarrow y$, connected through a natural language preference summary $z_i$. The key idea is to use a preference inference module directly with a downstream preference-alignment objective rather than a heuristic preprocessing step. The proposed advantage is "generator transferability", allowing pairing with any generator including frozen black-box APIs.

Both $f$ and $g$ are LLMs $(\pi_\phi​, \pi_\theta​)$ shared across users. The generator's DPO loss is modified by conditioning the policy on $z_i$ while keeping the reference conditioned only on $x$, yielding a "summary-augmented" DPO objective. The key observation is that the same $\mathcal{L}$ can supervise the inference model: since $z_i \sim \pi_\phi(\cdot \mid s_i)$ is a discrete sample,  $\mathcal{L}$ is non-differentiable in $\phi$, so they treat $-\ell_i$ as a scalar reward and optimize $\pi_\phi$​ with GRPO plus a KL divergence regularization term. Training is two-stage (RL on $\pi_\phi$​ with $\pi_\theta$​ frozen; optional DPO fine-tuning of $\pi_\theta$).

Experiments on ELIX, Review, Roleplay, and AlignX show substantial reductions in context length and generally improved personalization performance relative to prompting-based baselines. A central claim is that the learned summaries transfer well across eight frozen off-the-shelf generators including Claude-4 and GPT-4o-mini. There are also ablation studies including prompt templates, DPO temperature $\beta$, prompt-dependent summaries.

Strengths:
- The paper makes a valuable conceptual contribution. Instead of using user profiles for heuristic preprocessing signals, the idea is to make a user summary as the optimization objective of the downstream loss objective. The concrete approach to do this by making the summary-augmented DPO loss as the RL reward is promising.
- The generator transferability is a key strength of the approach relative to prior work. This is empirically validated, and provides evidence that the summaries encode meaningful and compressed preference information rather than some reward hacking for the training generator.
- Clear writing and limitations are presented upfront.
- Good ablation studies in experiments.

Weaknesses:
- Limited novelty of the optimization objective (essentially obtained by inserting a learned summary variable into an existing preference optimization objective)
- No variance estimates anywhere. No seeds reported, no Confidence intervels, no judge-order randomization. The statistical significance of the gains is unclear.
- Basic reproducibility facts are missing. The number of users N is never stated for any dataset, nor GRPO group size, rollout count, or RL steps.
- Missing baselines. Missing citations that are important baselines: RLPF — Reinforcement Learning from Prediction Feedback (Wu, Ning, Liu et al., AAAI 2025), PLUS — Preference Learning Using Summarization (Nam, Wan, Liu, Ahnn, Lian & Jaques, arXiv:2507.13579). Some of the citations should ideally be baselines, especially to clarify the role of optimization and prompt engineering: PAG / Profile-Augmented Generation, DPL, LaMP, Persona-Plug.

**Additional Comments:**

Questions for authors:
- Why is two-stage training preferable to alternating optimization?
- Is compression actually necessary?
- How stable are the learned summaries?

**Audience:**

Yes

**Audience Explanation:**

The question is timely and the design point is under-explored in LLM Personalization, which is one of the more active areas in post-training.
Generator-transferability is the paper's most distinctive contribution and is genuinely useful. The compression result is interesting independent of the personalization framing.

**Broader Impact Concerns:**

There might be concerns that are omitted in the broader impact statement. The techniques can be used to generate a human-readable natural-language profile of a user and then transmit that profile to arbitrary third-party, black-box, commercial generators which would likely need to be done carefully.

**Claims And Evidence:**

Yes

**Claims Explanation:**

While most of the claims are well-supported, I would like to point out there are some fixable instances of insufficient evidence which the authors should address.

Well-supported claims:
- POPI's optimized summaries are far shorter than the unoptimized reference model's on ELIX/Review/Roleplay, and this compression emerges without an explicit length penalty (Fig. 3).
- The summaries are not adversarial token strings (SUBTLEX).
- The framework is not tightly coupled to DPO (the IPO ablation, Table 6).
- The prompt-dependent preference summary buys little (Table 7), supporting the user-level scoping decision.
- Results are robust to the prompting template (Table 9), at least on average.

Could be supported better:
- Directly optimizing the preference-inference step is the effective lever. The summary content and summary length may be confounded.
- Generator-transferability is not supported on Roleplay. The low signal-to-noise in Roleplay's eight preference pairs is plausible but not tested.
- No uncertainty quantification, and the judge protocol is under-specified.
- "Minimizing $\mathcal{L}$ therefore simultaneously pursues two effect - it drives the generator to approximate the true preference distribution
under the summary, and it pushes the inference model $\pi_\phi$ to produce summaries carrying preference-relevant information beyond what the prompt already conveys". The loss decomposition is not as clean, since $z_i$ is drawn according to the inference model $\pi_\phi$, it affects the first KL term as well, and "reward hacking" is not completely eliminated.

**Requested Changes:**

1. Add a length-matched / format-matched Summary-Prompting baseline. This would isolate summary content from length/format.
2. Add uncertainty quantification (confidence intervals) and ensure that all the claims are statistically significant.
3. Substantiate the claim that "training becomes empirically robust to the specific choices of $\alpha$" by giving an $\alpha$-sweep at fixed $\beta$.
4. Report RL and selection details: GRPO group size, rollouts per prompt, total RL steps, sampling temperature.
5. [Optional] Investigate why generator fine-tuning hurts judged quality.
6. [Optional] Explain the Acc./Win-Rate divergence.

Also, address comments from above sections.